# Secure human action recognition by encrypted neural network inference

Miran Kim [1,2] ✉, Xiaoqian Jiang [3], Kristin Lauter[4], Elkhan Ismayilzada [5] & Shayan Shams[6] ✉

Advanced computer vision technology can provide near real-time home monitoring to support "aging in place" by detecting falls and symptoms related to seizures and stroke. Affordable webcams, together with cloud computing services (to run machine learning algorithms), can potentially bring significant social benefits. However, it has not been deployed in practice because of privacy concerns. In this paper, we propose a strategy that uses homomorphic encryption to resolve this dilemma, which guarantees information confidentiality while retaining action detection. Our protocol for secure inference can distinguish falls from activities of daily living with 86.21% sensitivity and 99.14% specificity, with an average inference latency of 1.2 seconds and 2.4 seconds on real-world test datasets using small and large neural nets, respectively. We show that our method enables a 613x speedup over the latency-optimized LoLa and achieves an average of 3.1x throughput increase in secure inference compared to the throughput-optimized nGraph-HE2.

Human action recognition has emerged as an important area of research in computer vision due to its numerous applications, such as in video surveillance, telemedicine, human–computer interaction, ambient assisted living, and robotics. In particular, application to telemedicine is becoming increasingly critical as changes in demographics, such as declining fertility rates and increasing longevity, have increased need for remote healthcare[1–3]. In many situations, elderly people who live alone do not receive immediate emergency assistance, and this failure may lead to serious injury or even death. Remote monitoring systems from healthcare providers can advance healthcare services, but healthcare providers cannot manually monitor hundreds of screens simultaneously. Uploading videos to cloud computing service providers (e.g., Amazon, Google, or Microsoft) and running recognition algorithms can be a promising way to solve this problem. Indeed, cloud-based services are becoming the mainstream in online marketplaces of digital services due to cost effectiveness and robustness. Edge devices have limited capacity to support miscellaneous and ever-growing types of digital services, so outsourcing of data and computation to the cloud is a natural choice.

A cloud service provider can provide real-time face detection and activity recognition (e.g., detecting behavioral pattern changes, emotion, falling, and seizure) on real-time video by adopting advanced artificial intelligence technologies. However, privacy concerns have become a critical hurdle in providing virtual remote healthcare to patients, especially in a cloud computing environment. Individual users do not want sensitive personal data to be shared with service providers. In this paper, we propose a secure service paradigm to reconcile the critical challenge by integrating machine learning (ML) techniques and fully homomorphic encryption (FHE). FHE enables us to perform high-throughput arithmetic operations on encrypted data without decrypting it, so the privacy-enhancing technology is considered to be a promising solution for secure outsourced computation[4,5]. Notably, a trusted home monitoring service was discussed as one of the applicable use scenarios in the Applications track at the 2020 HE Strategic Planning meeting and the related white paper was published[6].

In general, human action can be recognized from multiple modalities such as RGB images or video, depth, and body skeletons. Among these modalities, dynamic human skeletons, which represent 2D or 3D

[1]Department of Mathematics, Hanyang University, Seoul, Republic of Korea. [2]Department of Computer Science, Hanyang University, Seoul, Republic of Korea. [3]Center for Secure Artificial intelligence For hEalthcare (SAFE), School of Biomedical Informatics, University of Texas Health Science Center, Houston, TX, USA. [4]Meta AI Research, Seattle, WA, USA. [5]Department of Computer Science and Engineering, Ulsan National Institute of Science and Technology, Ulsan, Republic of Korea. [6]Department of Applied Data Science, San Jose State University, San Jose, CA, USA. ✉e-mail: miran@hanyang.ac.kr; Shayan.Shams@sjsu.edu

joint coordinates, have attracted more attention since they are robust against dynamic circumstances and are highly efficient in computation and storage[7–9]. In this work, we adopt a method that uses convolutional neural networks (CNN) for action recognition using the skeleton representation. The skeleton joints are easily captured by depth sensors or pose estimation algorithms[10–12]. Then the detected joint keypoints in the video streams are encrypted using the public key of FHE and sent to a cloud service provider. Then the cloud service runs the machine-learning algorithms on the encrypted joint points. After secure action recognition, the encrypted results are transmitted to a trusted party (e.g., a nursing station) who decrypts them and decides whether immediate intervention is necessary. Using encrypted joints as a uniform representation of the human body, our workflow supports multiple action recognition tasks concurrently as secure outsourcing tasks with cloud computing, which overcomes the scalability limitation of edge devices. This synergic combination of technologies can support monitoring of the elderly, while mitigating privacy concerns.

Theoretical progress has substantially reduced the time and memory requirements of secure computing[13–16], but adoption of it in real-world applications requires refinements in technology. A ciphertext in the FHE cryptosystem has an inherent error for security and multiplication operations bring about an increased noise level. Therefore, encryption parameters should be selected carefully to ensure both the security and correctness of a decryption procedure. Moreover, homomorphic operations result in different computational costs compared to plain computation, so a straightforward implementation (i.e., direct conversion of a plaintext computation into an encrypted domain) will be exceedingly slow. In particular, multiplication is a more costly operation than others. However, practical HE cryptosystems can only evaluate low-depth circuits, so for efficiency, it is imperative to balance multiplicative circuit depth and computation cost. Therefore, it is a non-trivial task to enable an efficient implementation of secure neural network inference with FHE.

In this paper, we present an FHE-compatible CNN architecture for skeleton-based action recognition, which is designed specially to be computed by a low-depth circuit with low-degree activation functions. Based on the proposed neural networks, we design a framework, named Homomorphically Encrypted Action Recognition (HEAR), which is a scalable and low-latency system to perform secure CNN inference as cloud outsourcing tasks without sacrificing accuracy of inference. We formulate a homomorphic convolution operation and propose an efficient evaluation strategy for the homomorphic convolution to exploit parallel computation on packed ciphertexts in a single instruction multiple data (SIMD) manner. We use the ciphertext packing technique to represent multiple nodes of layers as the same ciphertext while maintaining the row-major layout of tensors throughout the whole evaluation process. As a result, the secure inference solution avoids another level of complexity for switching back-and-forth between different data layouts over encryption. Additionally, the intensive use of both space and SIMD computation accelerates secure inference and reduces memory usage significantly. We demonstrate the effectiveness of our secure inference system on three benchmark datasets. HEAR enables a single prediction in 7.1 s on average over a 2D CNN model for action recognition tasks, while achieving 86.21% sensitivity and 99.14% specificity in detecting falls. Our elaborated and fine-tuned solution of Fast-HEAR can evaluate the same neural network in 2.4 s using only a few gigabytes of RAM while maintaining the same sensitivity and specificity in fall detection as HEAR. We also show that the proposed solutions achieve state-of-the-art latency and throughput of action inference over previous methods for secure neural network inference.

## Results
### Overview of HEAR
In the cloud-based action recognition system, FHE serves as a bridge to convert intrusive video monitoring into trustworthy services (Fig. 1). The HEAR system entails three parties: the monitoring service provider (e.g., a nursing station), end-users (data providers), and a cloud service provider. In our paradigm, we assume that model providers train a neural network with the cleartext data, and then offer the trained model to the public cloud. An end-user wants to be provided with privacy-preserving monitoring services while ensuring data confidentiality, so the user encrypts the data by using the public key of FHE and provides the encrypted data to the cloud server. The cloud server provides an online prediction service to data owners who uploaded their encrypted data by making predictions on encrypted data without decrypting them. After secure action recognition, the encrypted result is transmitted to the monitoring service provider, who decrypts it and decides how to respond to specific events. As described in Threat and Security Model in the "Methods" section, the security of the HE cryptosystem ensures that HEAR is secure against an honest-but-curious adversary.

### Innovation of HEAR
The conventional CNN architectures stack a few convolutional layers while periodically inserting a pooling layer between the convolutional

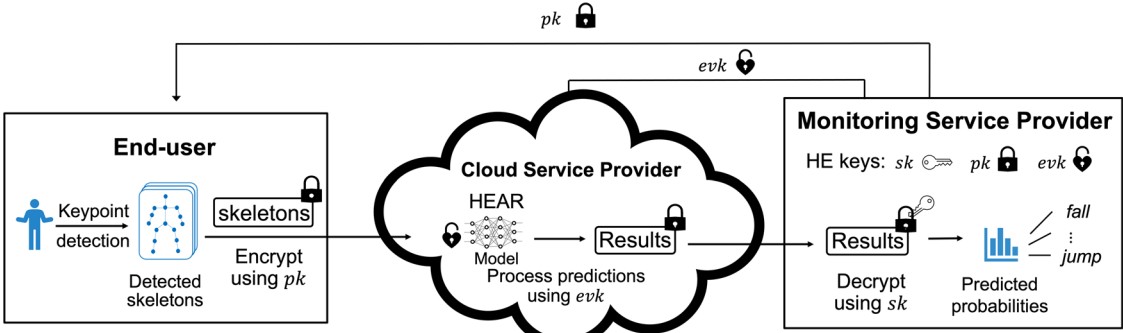

**Fig. 1 | A workflow of our cloud-based action recognition protocol.** At the beginning of the protocol, the monitoring service provider generates the cryptographic keys: (i) the secret key *sk* is used for decryption of ciphertexts; (ii) the public key *pk* is used for data encryption; and (iii) the evaluation keys *evk* are used for homomorphic computations (e.g., ciphertext-ciphertext multiplications or ciphertext rotations). The public key is transmitted securely to the end-users and the evaluation keys are transmitted securely to the cloud service provider. The cloud server is where encrypted data are processed while in encrypted form, so it has only access to the evaluation key for homomorphic computation. Video recordings by stationary video cameras are used to generate skeleton joints, which are encrypted using the public key of the underlying HE cryptosystem. The encrypted skeletons are fed to the cloud service provider. The cloud processes predictions on encrypted data and sends the encrypted classification results to the monitoring service provider. Finally, the nursing station of the monitoring service decrypts the results and responds to any alerts. For example, immediate intervention is necessary when a fall or seizure is detected.

**a**

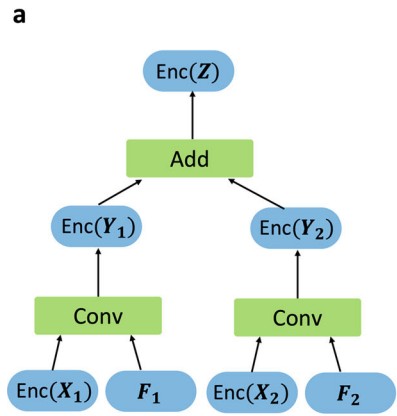

**b**

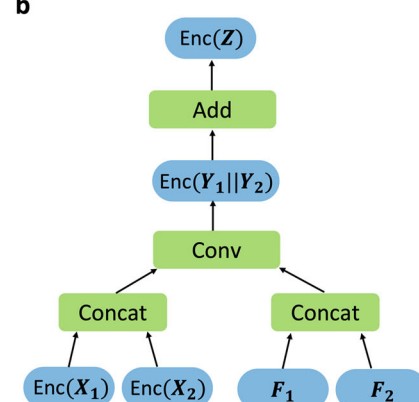

**Fig. 2 | Multi-channel ordinary homomorphic convolution and fast homomorphic convolution.** We denote by Enc(·) an encryption function. Given an input tensor **X** with two channels $X_1$ and $X_2$, $Y_i$ denotes an output by the single-input single-output convolution operation on the channel $X_i$ with the convolution kernel $F_i$. **Z** indicates a generated output channel by the convolution on the input **X**, which can be computed as $Z = Y_1 + Y_2$ in the clear. **a** Ordinary homomorphic convolution. Conv indicates a single-input single-output homomorphic convolution. Add indicates an ordinary homomorphic addition of two ciphertexts. **b** Fast homomorphic convolution. Concat indicates a concatenation over ciphertexts or plaintexts. Two punctured input ciphertexts Enc($X_1$) and Enc($X_2$) are fused to form one ciphertext by using concatenation over encryption. Then we perform homomorphic convolution operation on the packed ciphertext Enc($X_1||X_2$) by using the concatenated kernels ($F_1||F_2$), to yield a ciphertext that encrypts the intermediate results of ($Y_1||Y_2$). At the end, we perform a homomorphic addition of the values located in different entries of a plaintext vector, denoted by Add, which requires homomorphic rotations and additions over encryption. As a result, it yields a ciphertext that encrypts the output channel **Z**.

layers. Practical FHE schemes enable multiple values to be encrypted in a single ciphertext and perform computations on encrypted vectors in a SIMD manner[17], so an average-pooling operation can be implemented by a mean aggregation of adjacent entries of encrypted vectors by using homomorphic slot rotations, yielding ciphertexts with valid values stored sparsely. However, a decryption of intermediate results is not allowed during secure outsourced computations, so the sparsely packed ciphertexts are passed to the next convolutional layer. A straightforward method is to perform the ordinary homomorphic convolution on each sparsely packed ciphertext (Fig. 2a).

Here, we investigate the sparsity of ciphertexts to increase the efficiency of implementation of homomorphic convolution operations (Fig. 2b). We first formulate the homomorphic evaluation algorithm for multi-channel convolution operations as FHE-compatible operations on packed ciphertexts (e.g., SIMD addition, SIMD multiplication, and slot rotation). To maximize SIMD parallelism of computation, we extensively use entries that have non-valid values of ciphertexts. We put together as many sparsely packed ciphertexts of output channels from the previous pooling layer as possible into a single ciphertext while interlacing them with each other. As each kernel is applied onto an input channel, we perform simultaneous homomorphic convolution operations on the packed ciphertext, by using the concatenation of the corresponding kernels. The intermediate convolution results are involved together in the resulting ciphertext (i.e., the values are located in different entries in the corresponding plaintext vector), which is in turn summed together across plaintext slots to get the final output channel. The fast homomorphic convolution method incurs an additional cost to incorporate values at distinct ciphertexts before the ordinary convolution, and to aggregate values located in different slots after the ordinary convolution. However, the computational complexity of the convolution step is reduced by a predetermined factor compared to the naive homomorphic convolution, which allows greater efficiency. As a result, the whole process of the homomorphic convolution operations can be expressed as FHE-compatible operations on packed ciphertexts, which leads to a substantial speedup, especially in wide convolutional networks. This method has another advantage, in that it substantially reduces the amount of memory required for encoding model parameters as plaintext polynomials compared to the straightforward approach. Additionally, it maintains the row-major layout of tensors throughout the computation, thereby avoiding another level of computation for switching back-and-forth between different data layouts. Furthermore, we introduce a range of algorithmic and cryptographic optimizations tailored to increase the speed and reduce the memory usage of the secure neural network inference from the approximate HE cryptosystem (Cheon-Kim-Kimg-Song, CKKS[15]). To reduce the computational cost, we reformulate homomorphic convolutional operations by using the properties of ciphertext rotation operation. We also present the level-aware encoding strategy that represents the weight parameters as plaintext polynomials with small-sized coefficients enough to support required computations (i.e., having the minimum computational level budget). These innovations allow a speedup by an order of magnitude to encode model parameters as plaintexts, and enable a drastic reduction in the number of the encoded plaintexts.

### Dataset

Our dataset contains two categories of data: (i) Activities of daily living (ADLs) were selected from the J-HMDB dataset[18]. The selected action classes are clap, jump, pick, pour, run, sit, stand, walk, and wave. (ii) The fall action class was created by the UR Fall Detection dataset (URFD)[19] and the Multiple cameras fall dataset (Multicam)[20]. OpenCV (version 3.4.1) was used for image processing. We used the pytorch (version 1.3) implementation of the Deep High-Resolution network (HRNet, https://github.com/leoxiaobin/deep-high-resolution-net.pytorch)[12] pretrained with the MPII Human Pose dataset[21] to detect keypoint locations. The network outputs 15 joint locations of each frame: ankles, knees, hips, shoulders, wrists, elbows, upper neck, and head top. For each dataset, the skeleton joints are first arranged as a 3D tensor of size $2 \times 32 \times 15$ by concatenating the detected joint locations from 32 frames of the generated clips. The transformed samples from the three datasets are merged for analysis, and the merged dataset is split randomly into training and testing sets that contain 70% (84 falls and 1346 non-falls) and 30% (29 falls and 579 non-falls), respectively. We note that it takes around 94 ms to detect 15 joint locations for each frame on a V100 GPU. So, it takes about 3.008 s to generate a 3D tensor from extracted skeleton joints of 32 frames.

### Network architecture for action recognition

Our plain action recognition network was inspired by the design of Du et al.[22] to capture spatial-temporal information. The network consists

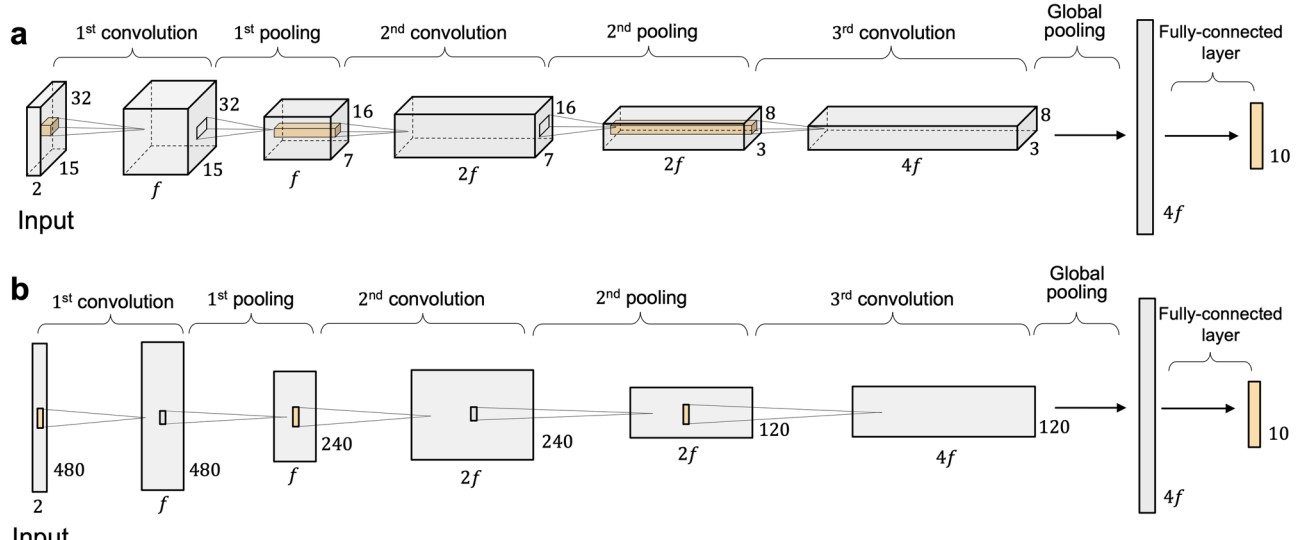

**Fig. 3 | Convolutional neural network architectures for our action recognition.** $f$ denotes the number of filters in the first convolutional layer, and we used $f \in \{64, 128\}$ in the experiment. **a** The convolutional neural network with the 2D convolutions. **b** The convolutional neural network with the 1D convolutions.

of three convolutional layers, which are each followed by a batch normalization (BN), an activation layer, and a downsampling layer. The network ends with a fully connected (FC) layer and softmax. We consider two CNN models depending on the shape of input neurons and the movement of kernels for a convolution operation. Each neuron in the 2D-CNN models contains two-dimensional planes for input, and the network consists of 2D convolutions in which the kernel slides along two dimensions over the data (Fig. 3a). In the 1D-CNN models, 2D matrices for kernels and feature maps are replaced with 1D arrays, and the kernel slides along one dimension over the data (Fig. 3b). The convolutional layers have a filter size of 3 × 3 (2D-CNN) or 3 (1D-CNN), a stride of 1, and the same padding. We follow the design rule of ResNet[23] such that if the feature map size is halved, the number of filters in the convolutional layers is increased to doubled. In our experiments, we study one small net and one large net: CNN-64 and CNN-128, where 64 and 128 represent the number of filters in the first convolutional layer, respectively. We replace the ReLU activation with a quadratic polynomial, and adjust the coefficients during the training phase. The downsampling is performed by average-pooling over window size of 2 × 2 or 2, with a stride of 2, or a global average pooling at the end.

### Homomorphic convolution benchmark

Given a load number $n_P$ as the number of ciphertexts to fit into a single ciphertext in the preprocessing step, we get a speedup of up to $n_P$ for the convolution operation (Table 1). Therefore, we may offer more performance benefits if we assemble as many as ciphertexts within the parameter limit and perform the homomorphic convolution on the packed ciphertext. To be specific, we get the load number $n_P$ of the $t$-th convolutional layer as $2^{t-1}$ in 1D-CNN and $2^{2(t-1)}$ in 2D-CNN. For instance, the third convolutional layer in the 2D-CNN-128 network has a load number of $n_P = 16$ in the Fast-HEAR system, and achieves a significant speedup over HEAR. However, Fast-HEAR requires additional computational costs for pre/post-processing procedures, so the speedup for the whole convolutional layer is slightly smaller than the load number.

### Time requirement of secure action recognition

To demonstrate the scalability and practicability of our secure action recognition protocol, we performed a detailed analysis of running time requirement for the HEAR and Fast-HEAR systems over various CNN models. We divide the process into five steps: key generation, encoding of weight parameters, data encryption, secure inference, and

decryption. (i) Key generation: Fast-HEAR requires one additional level of plaintext-ciphertext multiplication for each preprocessing step than HEAR, so Fast-HEAR uses slightly larger HE keys than HEAR, and thereby incurs 38–62% increase in runtime for key generation (Fig. 4a). The FHE cryptosystem requires public rotation keys specified by rotation amounts for ciphertext rotations. The large increase in runtime between HEAR and Fast-HEAR from 1D models to 2D models is due to an increasing number of rotation keys required for the preprocessing and postprocessing steps. (ii) Encoding of weight parameters: In the Fast-HEAR system, the weight parameters are encoded as plaintext polynomials more compactly together than in HEAR, to align with packed ciphertexts; this difference shows a considerable reduction in time and memory usage when a large load number $n_P$ is used. For example, Fast-HEAR has the largest parameters $n_P$ in the 2D-CNN-128 network, so Fast-HEAR shows the largest speedup of 7x over HEAR (Fig. 4b). (iii) Encryption: Both systems takes 1.34–1.54 s to encrypt 608 samples of the test set, yielding an amortized rate of 22–25 ms per sample. (iv) Secure inference: The intensive use of SIMD computation in Fast-HEAR speeds up the process of secure inference (Fig. 4c). The average speedup of Fast-HEAR over HEAR on the test set using the 2D-CNN-128 network inference is 3x (7.073 s vs 2.419 s). In particular, Fast-HEAR achieves a substantial improvement of 2D CNN inference over HEAR, because 2D CNN uses a larger $n_P$ than 1D CNN, even for the same number of filters. (v) Decryption: After the evaluation, the cloud server outputs a single ciphertext of the predicted results; the decryption takes 1.6 ms on average.

### Memory requirements of secure action recognition

The Fast-HEAR system offers the substantial memory benefit for storing model parameters. Fast-HEAR uses 35% and 15% as much space as HEAR to encode the weight parameters on the 1D-CNN and 2D-CNN models, respectively (Fig. 4d, e). This speedup occurs because the filters are packed more tightly in Fast-HEAR system than in HEAR. Furthermore, Fast-HEAR shows better memory management in homomorphic computation by using 47%–64% as much space as HEAR.

### Communication cost

A freshly encrypted input tensor of the network has ~1.4 –1.6 MB from the user to the cloud server. The current protocol can make 3600/2.4 ≈ 1500 predictions per hour using the 2D-CNN-128 network on a

**Table 1 | Homomorphic convolution microbenchmark in our action recognition network**

| Network | Output | | | #(Ciphertexts) | | | Ordinary Convolution (milliseconds) | Fast convolution (milliseconds) | | | | Speedup From Fast-HEAR |
|---|---|---|---|---|---|---|---|---|---|---|---|---|
| | Layer | Map Size | #Filters | Input $n_{in}$ | Output $n_{out}$ | Packed $n_P$ | | Steps Pre | Conv | Post | Total | |
| [a]1D-CNN-64 | conv2 | 240 | 128 | 4 | 8 | 2 | 409±30 | 65±7 | 283±19 | 42±6 | 390±22 | 1.1 |
| | conv3 | 120 | 256 | 8 | 16 | 4 | 422±53 | 49±10 | 156±10 | 37±5 | 242±17 | 1.7 |
| 1D-CNN-128 | conv2 | 240 | 256 | 8 | 16 | 2 | 794±108 | 68±7 | 560±30 | 47±3 | 674±30 | 1.2 |
| | conv3 | 120 | 512 | 16 | 32 | 4 | 1416±215 | 47±10 | 408±29 | 59±6 | 514±32 | 2.8 |
| 2D-CNN-64 | conv2 | (16,7) | 128 | 4 | 8 | 4 | 985±136 | 70±12 | 550±31 | 74±5 | 694±35 | 1.4 |
| | conv3 | (8,3) | 256 | 8 | 16 | 8 | 1145±217 | 47±11 | 288±20 | 50±5 | 385±23 | 3.0 |
| 2D-CNN-128 | conv2 | (16,7) | 256 | 8 | 16 | 4 | 2173±414 | 75±12 | 856±51 | 79±4 | 1010±54 | **2.2** |
| | conv3 | (8,3) | 512 | 16 | 32 | 16 | 4251±766 | 49±10 | 507±31 | 104±6 | 660±35 | 6.4 |

In the second column, *convi* indicates the *i*-th convolutional layer in the network. The three columns for #(Ciphertexts) correspond the number $n_{in}$ of input ciphertexts, the number $n_{out}$ of output ciphertexts, and the number $n_p$ of input ciphertexts to fit into a single ciphertext for the fast homomorphic convolution in the HEAR system. The timing results reported are mean ± standard deviation (s.d.) from $n = 608$ independent samples on the test set. The column for Ordinary convolution gives timing for the ordinary homomorphic convolution over assembled ciphertexts. Three columns for Fast convolution correspond to the preprocessing step (fusing different ciphertext into a single ciphertext), the ordinary multi-channel homomorphic convolutions over assembled ciphertexts, and the postprocessing step (accumulation of intermediate convolution results). The total execution time of these procedures is given in the following column. The last column gives a speedup of the fast homomorphic convolution over the ordinary homomorphic convolution.

single server. The encrypted prediction result is ~0.13 MB, so servers are sufficient to support the $0.13 \times 1500 \approx 195$ MB bandwidth requirement for ciphertexts loads to the monitoring service provider per hour.

## Classification performance

To examine the classification performance of fall detection, we used the typical performance metrics such as classification accuracy, sensitivity, specificity, precision, and F1-score. We expected that our selection of parameter sets would offer a trade-off between evaluation performance and output precision of HEAR and Fast-HEAR. Surprisingly, both secure inference solutions achieved the same performance on the test set as the unencrypted inference except for the classification accuracy (Fig. 4f). Our systems can distinguish falls from ADLs with 86.21% sensitivity, 99.14% specificity, and 84.75% F1-score on all networks. When the large neural nets are evaluated (e.g., CNN-128), the values are slightly affected by errors from homomorphic computations, and therefore show an accuracy degradation of 0.16–0.17%. These results indicate that the proposed secure methods show perfect data protection at the cost of a slight loss in classification accuracy. Notably, the promising point is that our solutions can distinguish between falls and non-falls just as well as the unencrypted inference can (Fig. 4g, h).

## Comparison to prior work

CryptoNets[24] was the first protocol for enabling secure neural network inference on the MNIST dataset[25]. Their protocol is to encrypt each node in the network as distinct ciphertexts and emulate unencrypted computation of neural network inference in the normal way while making predictions on thousands of inputs at a time. The follow-up studies of nGraph-HE[26] and nGraph-HE2[27] significantly improved the inference throughput by using scheme-dependent cryptographic optimizations of underlying homomorphic operations such as plaintext-ciphertext addition and multiplication. However, they have high inference latency even for a single prediction and can lead to memory problems when applied to large-scale neural networks.

The most relevant method is LoLa[28], which uses the ciphertext packing method to represent multiple values from network nodes as the same ciphertext. In LoLa, a convolutional layer is expressed either as a restricted linear operation by flattening the kernels to a single dimension, or as a product of a large weight matrix and an encrypted data vector. In particular, the matrix-vector product is computed simply by a series of dot-products between each row of the matrix and the data vector, giving the output of each filter at each location. However, these simplifications lead to a substantial number of homomorphic operations over large-dimensional inputs, which is the case for wide networks. We refer to the "Methods" section for a theoretical comparison of computational costs of homomorphic convolutions in nGraph-HE2, LoLa, and Fast-HEAR.

We provide the runtime for homomorphic evaluation of nGraph-HE2 and LoLa over various neural network models (Fig. 5a). Specifically, nGraph-HE2 performs 608 predictions simultaneously in 1.2 h using the 2D-CNN-128 network; of this time, the second and third convolutional layers consume 47 min and 17 min, respectively (Fig. 5b). LoLa performs a single prediction in 15.4 min; of this time, the second and third convolutional layers consume 13 min and 2.4 min, respectively. Fast-HEAR achieves a 3.1 times increase in secure inference throughput on average compared to the throughput-optimized nGraph-HE2, and it is on average 613 times faster than the latency-optimized LoLa in secure inference (Fig. 5c). We note that nGraph-HE2 takes on average 11.8 s to encrypt 608 samples of the test set into a single ciphertext over the 2D-CNN-128 network from $n = 10$ independent experiments, yielding an amortized rate of 19 ms per sample. LoLa takes on average 74 ms to encrypt a single sample over the same network from $n = 10$ independent experiments. We measured average

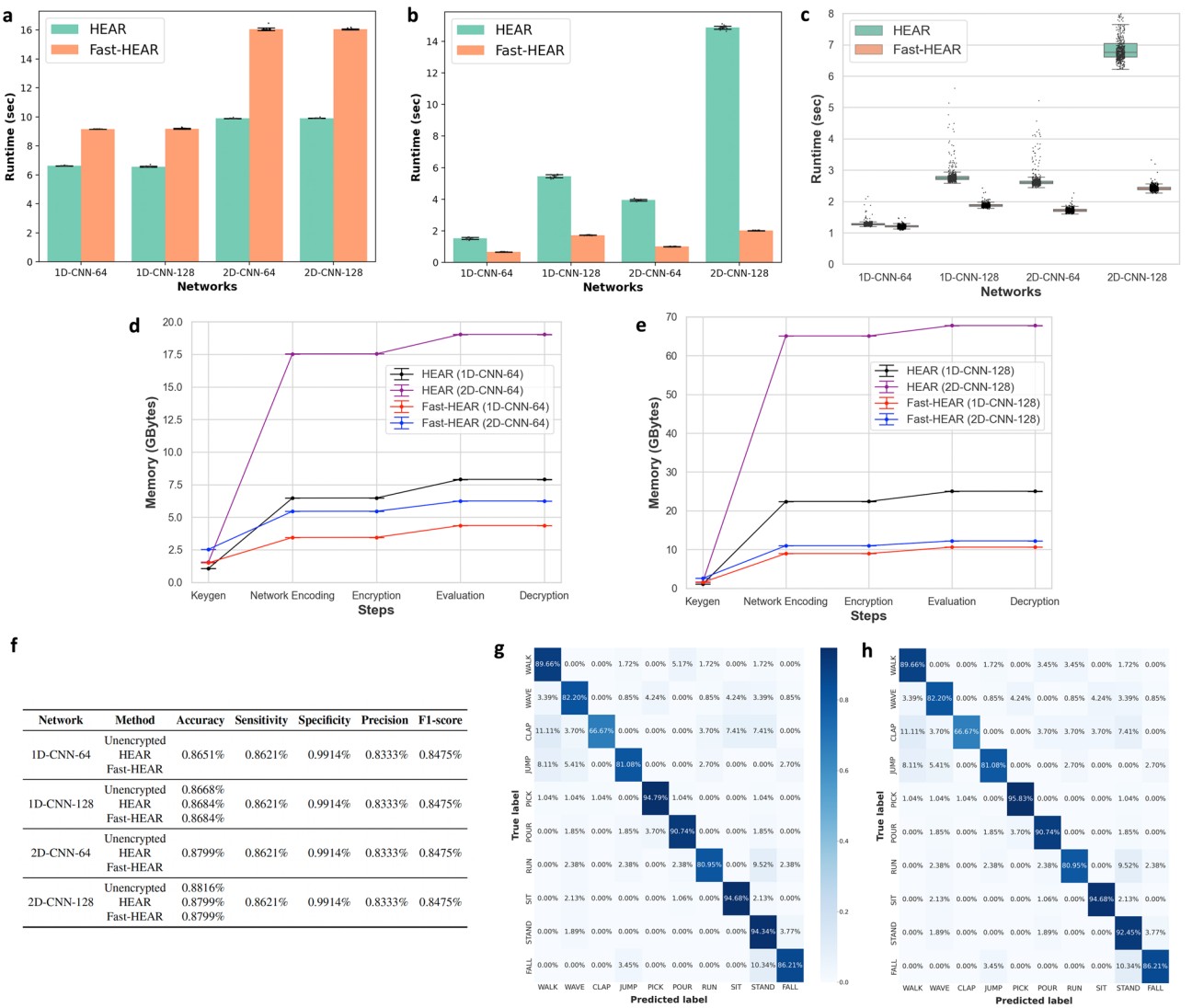

**Fig. 4 | Experimental results of secure action recognition inference over various CNN networks. a** Running time required to generate all required cryptographic keys for secure computation. Data are presented as mean ± s.d. from $n = 20$ independent experiments. **b** Running time for encoding the weight parameters. Data are presented as mean ± s.d. from $n = 20$ independent experiments. **c** Average running time for secure inference on $n = 608$ independent samples from the test set over various neural network models. Boxplot displays the median values with the first and third quartiles, and the whiskers boundaries extend to the largest and smallest data values no more than 1.5 times the interquartile range (IQR) from the corresponding hinge. **d**, **e** Average peak memory usage during execution of homomorphic computation on the CNN-64 network models (**d**) and CNN-128 network models (**e**). Data are presented as mean ± s.d. of $n = 20$ independent experiments. **f** Classification performance comparisons of the unencrypted and encrypted models on the test set. **g**, **h** Confusion matrices of the unencrypted computation (**g**) and Fast-HEAR system (**h**) on the test set over the 2D-CNN-128 network.

memory usage during secure inference (Fig. 5d, e). The implementation of nGraph-HE2 consumes 98.5%–99.5% of the memory utilization during homomorphic computation (376 GB on average). In particular, the memory usage for the evaluation step in nGraph-HE2 showed a similar tendency to increasing numbers of intermediate channels during an unencrypted computation. In contrast, the implementation of LoLa consumes 98.2%–99.7% of the memory utilization during parameter encoding (547 GB on average). As a result, Fast-HEAR uses 97.8%–98.5% less space than nGraph-HE2 and LoLa, and therefore uses a significantly less memory usage than they do. We remark that nGraph-HE2 and LoLa have the same multiplicative circuit depths as HEAR and Fast-HEAR, respectively. nGraph-HE2 and HEAR emulate an unencrypted inference process using different network node encryption methods, so they have the same depth for secure inference. On the other hand, Fast-HEAR requires one more plaintext-ciphertext multiplication to put sparsely packed ciphertexts into a single ciphertext in the 2nd and 3rd fast convolution than HEAR. Similarly,

LoLa requires one more plaintext-ciphertext multiplication after the matrix-vector product (i.e., 2nd and 3rd convolutions) as the scattered results of the product are packed into the same ciphertext and performed together by the subsequent activation. As a result, we set the same encryption parameters of nGraph-HE2 and LoLa as HEAR and Fast-HEAR, respectively. As errors from homomorphic computations are determined primarily by encryption parameters, nGraph-HE2 and LoLa achieved the same classification performance on the test set as our methods (Fig. 5f).

Other approaches are available for privacy-preserving deep learning prediction that uses multi-party computation (MPC) and their combinations with HE[29,30]. These methods provide good latency but assume the tolerance of intensive communication overhead, which is not feasible in practice, because the number of bits that the parties need to exchange during the MPC protocol is proportional to the number of nodes in the neural network. Most of all, the systems are interactive, so all participating parties should join the computation,

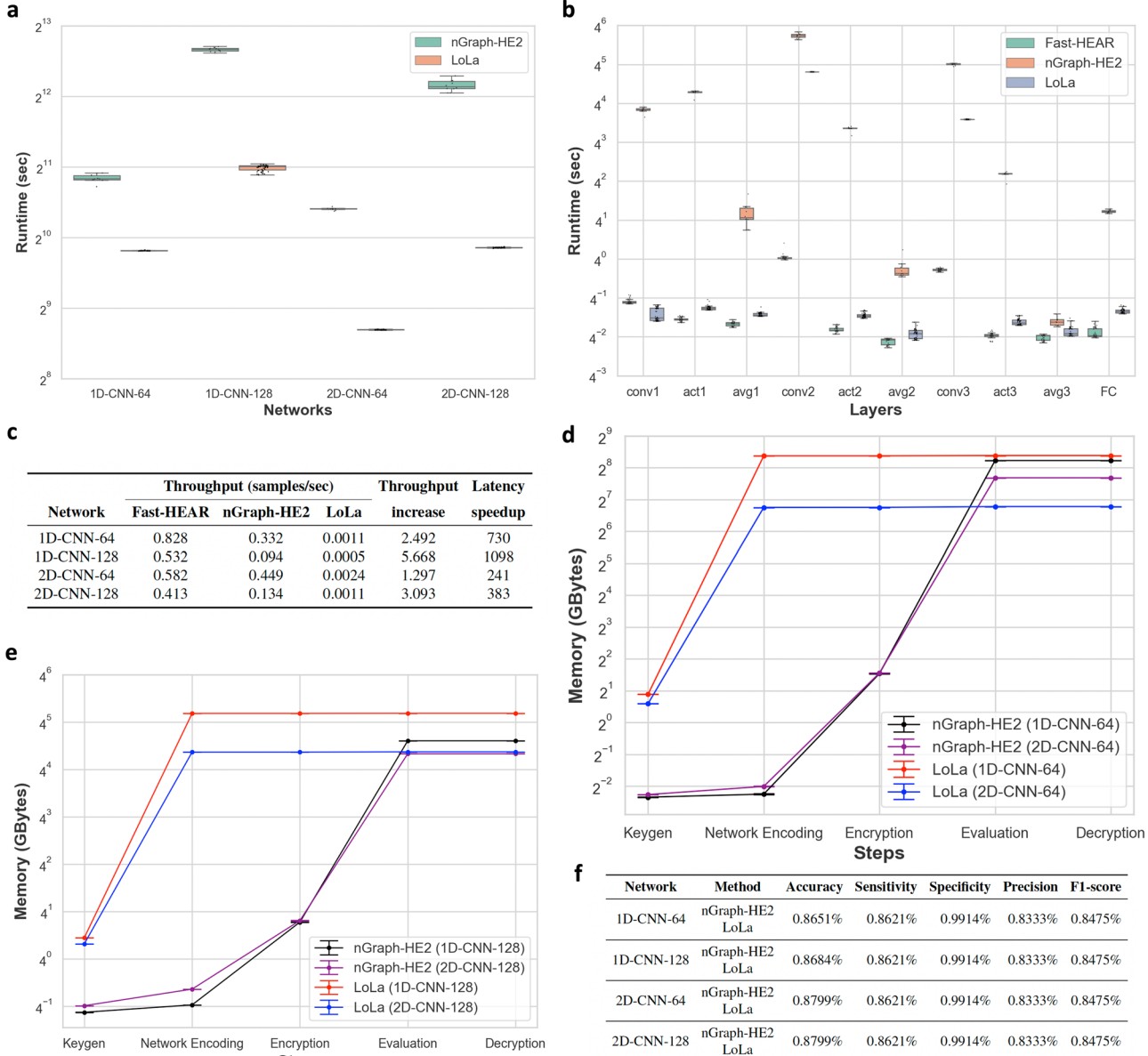

**Fig. 5 | Comparison with state-of-the-art methods. a, b** Boxplots display the median values with the first and third quartiles, and the whiskers boundaries represent the largest and smallest data values no more than 1.5 times the IQR from the corresponding hinge. Average running time for secure inference of nGraph-HE2 (from $n = 10$ independent experiments) and LoLa (from $n = 50$ independent samples) over various neural network models (**a**). Detailed average running time for each step in secure inference of Fast-HEAR (from $n = 50$ independent samples), nGraph-HE2 (from $n = 10$ independent experiments), and LoLa (from $n = 50$ independent samples) over the 2D-CNN-128 network (**b**). **c** Performance comparison including secure inference throughput (samples per second). The fifth column indicates the throughput increase from Fast-HEAR over nGraph-HE2. The last column indicates the latency speedup from Fast-HEAR over LoLa. **d, e** Average peak memory usage during execution of homomorphic computation on the CNN-64 network models (**d**) and CNN-128 network models (**e**). Data are presented as mean ± s.d. from $n = 10$ independent experiments. **f** Classification performance comparisons of the models on the test set.

and this requirement demands an additional complicated setup. Therefore, data providers should stay online during the entire protocol execution, and it is difficult to operate in reality. In our case, multiple values from input are encrypted as a single ciphertext and it is enough to be transmitted once before secure computation. The communication cost is proportional to the number of inputs, so our solution is asymptotically more efficient in communication than those other approaches. Contrary to MPC-based approaches, a service provider performs a large amount of work, and a client does not need to be involved in the computation. Additionally, the hybrid protocols require decryption of homomorphically encrypted ciphertexts after linear computation, which can leak information about data. Instead, our method provides end-to-end encryption and is allowed to decrypt only the predicted result, so it does not leak any information about data.

## Discussion

Homomorphic encryption has recently attracted much attention in the application of privacy-preserving Machine Learning as a Service (MLaaS). In this paper, we address the real-world challenge in privacy-preserving human action recognition by presenting a scalable and low-latency HE-based system for secure neural network inference. Our solution shows highly promising results for enhancing privacy-preserving healthcare monitoring services for aging in place in a cost-effective and reliable manner, which can have abundant social and health value.

Our study was enabled by the synergistic combination of machine-learning technologies and cryptographic development. Although significant progress in the theory and practice of FHE has been made towards improving efficiency in recent years, FHE-based approaches are believed to have a key bottleneck to achieving practical performance and the cryptographic protocol is still regarded as theoretical. However, theoretical breakthroughs in the HE literature and a strong effort of the FHE community[31] have enabled massive progress and offered excellent potential for secure computation in a wide range of real-world applications such as machine learning[24,27,28,32], biomedical analysis[33–35], private set intersection[36], and private information retrieval[37].

Notably, iDASH (integrating Data for Analysis, Anonymization, Sharing)[38] has hosted a secure genome analysis competition over the last decade, and practical yet rigorous solutions to real-world biomedical privacy challenges are being developed. Recently, FHE-based machine-learning approaches[39,40] have demonstrated the feasibility and scalability of privacy-preserving genomic data analysis. We hope that our study can provide a reference for the development of FHE-based secure approaches.

In reality, different health-related events bear out different weights for each individual. Clinical applications (i.e., stroke rehabilitation, Alzheimer's disease monitoring) would benefit from different action recognition tasks measured at different frequencies. Fall recognition is one application of remote healthcare. A client (especially an elderly one who has comorbidities) may subscribe to multiple tasks; if so, they must be deployed across multiple neural networks and managed by the backend algorithms simultaneously. In a cloud-based outsourced scenario, the user only needs to encrypt the data once before outsourcing them, and the encrypted data can be used for different tasks. This characteristic eliminates the need to reprogram the end devices whenever the model providers tweak neural network models, so the overhead of end-users is significantly reduced. As a result, our solution can support multiple concurrent and heterogeneous tasks in elastic cloud computing with mitigated privacy risks for end-users. Therefore, by secure outsourcing with HE, the architecture allows us to build a secure and privacy-preserving ecosystem between algorithm developers and data owners.

The presented secure inference method is based on wide CNN models. Non-detected fall may lead to a death, so further improvement is required to improve the sensitivity of a classifier by increasing the depth or width of the network, or using more complex network models. Nevertheless, we expect that the proposed evaluation approach can be used for such networks to provide accurate inference. However, depending on applications, the algorithm developers or providers might not want to disclose their intellectual properties. For example, a company trained a machine-learning model on sensitive private data from their customers. To decrease the risk of data being intercepted, damaged, or stolen, the clouds are provisioned with encrypted prediction models to use as a classifier. We are foreseeing that it can be addressed by adapting our secure inference method. Another limitation is that the current CNN computation was manually designed, heavily optimized, and carefully implemented by using the structure of networks. An avenue for future research is to build a deep learning computation protocol that exploits our findings, automatically generates homomorphic tensor operations, and optimizes the end-to-end performance.

## Methods

### Threat model

We consider the following threat models. First, we assume that all parties are semi-honest (i.e., honest but curious); that is, they follow the protocol and execute all steps correctly. The underlying HE scheme is indistinguishable against chosen-plaintext attack (IND-CPA) under the Ring Learning with Errors assumption[41]. All computations on the server are processed in encrypted form, so the server does not learn anything about the user's input due to the IND-CPA security of HE. Therefore, we can ensure the confidentiality of data against the cloud service provider. Second, secure authenticated channels are required between end-users and cloud and between cloud and monitoring service providers to prevent an attacker from tampering with an encrypted user's data or impersonating the cloud or the monitoring service provider. Third, we assume that the monitoring service provider is not allowed to collude with the cloud server. The cloud can access the decrypted skeleton joints if they share data. Finally, we remark that the CKKS scheme is secure against the key-retrieval attack if plaintext results of decryption are revealed only to the secret-key owner[42]. The decrypted results from the monitoring service provider are not shared with any external party, so our protocol is secure against the key-retrieval attack.

### Notation

The binary logarithm will be simply denoted by $\log(\cdot)$, and $v[i]$ indicates the $i$-th entry of the vector $\mathbf{v}$. If two matrices $\mathbf{A}_1$ and $\mathbf{A}_2$ have the same number of rows, $(\mathbf{A}_1|\mathbf{A}_2)$ denotes a matrix formed by horizontal concatenation. We use a row-major ordering map to transform a matrix in $\mathbb{R}^{d_1 \times d_2}$ into a a vector of dimension $n = d_1 d_2$. More specifically, for a matrix $\mathbf{A} = (a_{ij}) \in \mathbb{R}^{d_1 \times d_2}$, we define a bijective map $\text{vec} : \mathbb{R}^{d_1 \times d_2} \to \mathbb{R}^n$ by $\text{vec}(\mathbf{A}) = (a_{11}, a_{12}, \ldots, a_{1d_2}, \ldots, a_{d_1 1}, a_{d_1 2}, \ldots, a_{d_1 d_2})$. The vectorization can be extended to tensors. A tensor $\mathbf{A} \in \mathbb{R}^{d_1 \times d_2 \times d_3}$ is simply interpreted as a vector in $\mathbb{R}^{d_1 \cdot d_2 \cdot d_3}$ by $\text{vec}(\mathbf{A}) = (\text{vec}(\mathbf{A}_1)|\text{vec}(\mathbf{A}_2)|\ldots|\text{vec}(\mathbf{A}_{d_1}))$, where $\mathbf{A}_\ell \in \mathbb{R}^{d_2 \times d_3}$ is a matrix obtained by taking an index of $\ell$ in the outermost dimension. The vectorization process matches a method for storing tensors in the row-major order (i.e., the inner-most dimension is contiguously stored).

### Single-channel homomorphic convolutions

We start with a simple convolution of a single-input $\mathbf{X}$ in $\mathbb{R}^{h \times w}$ with a single $(f_h \times f_w)$ filter and the stride parameters $(s_h, s_w)$. The output of a neuron in a convolutional layer can be computed as

$$z_{i,j} = \sum_{|u| \leq \lfloor f_h/2 \rfloor} \sum_{|v| \leq \lfloor f_w/2 \rfloor} x_{i \cdot s_h + u, j \cdot s_w + v} \cdot f_{u,v}, \tag{1}$$

where $x_{i,j}$ and $z_{i,j}$ the input and output of the neuron located in row $i$ and column $j$, and $f_{u,v}$ represents the weight located at row $u$ and column $v$. Assume that the input channel is encrypted as a single ciphertext in row-major order, i.e., it is converted into a 1D vector by vectorization of a matrix and the resulting plaintext vector is encrypted. As the convolution kernel slides along the input matrix, we perform a dot product of the kernel with the input at each sliding position. We can take advantage of SIMD computation to get convolution results at all the positions at a time. This can be achieved by simply computing $f_h \cdot f_w$ rotations of the encrypted input, multiplying each rotated ciphertext by a plaintext polynomial with the weights of the filter, and adding the resulting ciphertexts. To be specific, we have

$$x_{i \cdot s_h + u, j \cdot s_w + v} = \text{vec}(\mathbf{X})[(i \cdot s_h + u) \cdot w + j \cdot s_w + v] = \rho^{u \cdot w + v}(\text{vec}(\mathbf{X}))[i \cdot s_h \cdot w + j \cdot s_w] \tag{2}$$

where $\rho^\ell$ indicates a rotation operation to the left by $\ell$ positions, so Equation (1) can be expressed as follows:

$$z_{i,j} = \sum_{|u| \leq \lfloor f_h/2 \rfloor} \sum_{|v| \leq \lfloor f_w/2 \rfloor} (\rho^{u \cdot w + v}(\text{vec}(\mathbf{X}))[i \cdot s_h \cdot w + j \cdot s_w]) \cdot f_{u,v} \tag{3}$$

$$= \left( \sum_{|u| \leq \lfloor f_h/2 \rfloor} \sum_{|v| \leq \lfloor f_w/2 \rfloor} \rho^{u \cdot w + v}(\text{vec}(\mathbf{X})) \cdot f_{u,v} \right)[i \cdot s_h \cdot w + j \cdot s_w]. \tag{4}$$

 

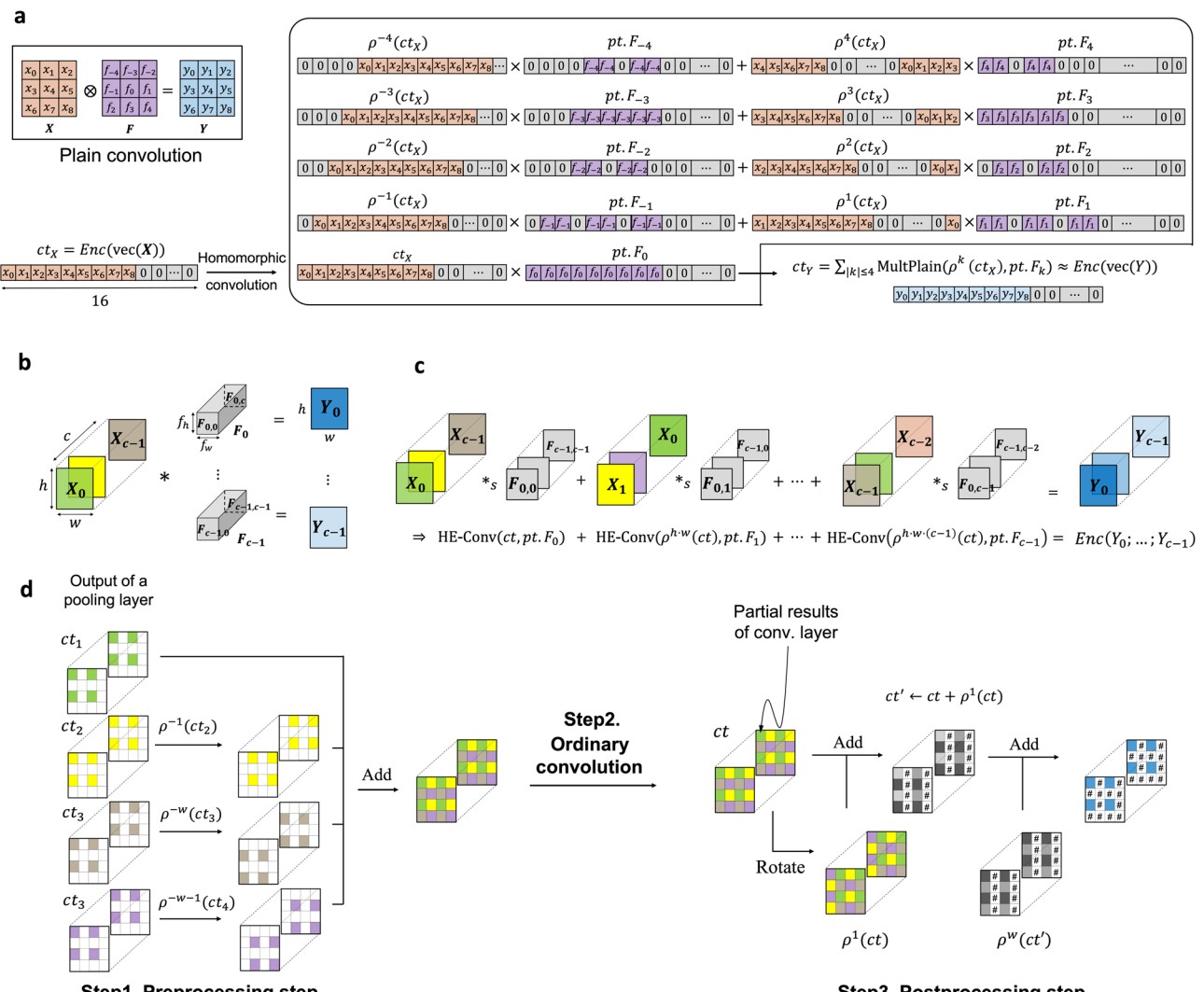

**Fig. 6 | Illustration of homomorphic 2D convolution operations.**
**a** Homomorphic evaluation of ordinary 2D convolution with a kernel size of 3 and a stride of 1. We denote by vec($\cdot$) a row-major vectorization, Enc($\cdot$) an encryption function, $\rho^\ell(\cdot)$ a ciphertext rotation to the left by $\ell$ positions, and MultPlain a plaintext-ciphertext multiplication. **b** Plain convolution algorithm of the feature maps $\{\mathbf{F_j} = (\mathbf{F_{jl}})_{l=0}^{c-1} \in \mathbb{R}^{c \times f_h \times f_w}\}_{0 \le j < c}$ on an input $\mathbf{X} = \{\mathbf{X_i} \in \mathbb{R}^{h \times w}\}_{0 \le i < c}$ and an output $\mathbf{Y} = \{\mathbf{Y_i} \in \mathbb{R}^{h \times w}\}_{0 \le i < c}$. Operation * indicates the multi-channel convolution. **c** Homomorphic convolution algorithm. The input $\mathbf{X}$ is given as a fully packed

ciphertext ct, and pt.$\mathbf{F_j} = \{\text{pt.F}_{\mathbf{j,u,v}}\}_{|u| \le \lfloor \mathbf{f_h}/2 \rfloor, |v| \le \lfloor \mathbf{f_w}/2 \rfloor}$ denotes a set of the plaintext polynomials of the kernels $\mathbf{F}_{0j}, \mathbf{F}_{1,j+1}, ..., \mathbf{F}_{c-1j+c-1}$. Operation $*_s$ indicates the parallelized ordinary convolution of a single-input channel with a single kernel over encryption, where HE $\cdot$ Conv(ct$_\mathbf{X}$, {pt.F$_{\mathbf{u,v}}$}) = $\sum_{|u| \le \lfloor \mathbf{f_h}/2 \rfloor} \sum_{|v| \le \lfloor \mathbf{f_w}/2 \rfloor}$ MultPlain($\rho^{\mathbf{u \cdot w} + \mathbf{v}}$(ct$_\mathbf{X}$), pt.F$_{\mathbf{u,v}}$). **d** Preprocessing and postprocessing procedures for fast homomorphic convolution. Colored entries in the preprocessing step are valid values as the output response maps of the pooling layer; entries marked # are non-valid values.

Accordingly, the simple convolution on an input ciphertext ct$_\mathbf{X}$ of the input $\mathbf{X}$ can be computed by

$$\text{HE} - \text{Conv}(\text{ct}_\mathbf{X}, \{\text{pt.F}_{u,v}\}) = \sum_{|u| \le \lfloor f_h/2 \rfloor} \sum_{|v| \le \lfloor f_w/2 \rfloor} \text{MultPlain}(\rho^{u \cdot w + v}(\text{ct}_\mathbf{X}), \text{pt.F}_{u,v}), \quad (5)$$

where pt. F$_{u,v}$ are plaintext polynomials that have the weights of the filters in appropriate locations, and MultPlain(ct, pt) denotes a multiplication of a plaintext pt to a ciphertext ct (Fig. 6a). It follows from Equation (4) that the $(i \cdot s_h \cdot w + j \cdot s_w)$-th entry of the resulting ciphertext is $z_{i,j}$. We remark that Equation (5) can be applied to the 1D convolution by taking a filter size $f_h \times f_w$ as $f_h = 1$.

**Multi-channel homomorphic convolutions**
The multi-channel convolution is represented as $c$ filter banks $\{\mathbf{F_j} \in \mathbb{R}^{c \times f_h \times f_w}\}$ on an input tensor $\mathbf{X} \in \mathbb{R}^{c \times h \times w}$ (Fig. 6b). For $0 \le \ell < c$, let $\mathbf{X}_\ell = \mathbf{X}_{\ell,:,:}$ be the matrix obtained by taking an index of $\ell$ in the outermost dimension. For the sake of brevity, we assume that the input

tensor $\mathbf{X}$ is given as a ciphertext ct representing its vectorization. Then, the homomorphic property yields

$$\text{Dec}(\rho^{h \cdot w \cdot \ell}(\text{ct})) \approx \rho^{h \cdot w \cdot \ell}(\text{vec}(\mathbf{X})) = (\text{vec}(\mathbf{X}_\ell)|\text{vec}(\mathbf{X}_{\ell+1})|\cdots|\text{vec}(\mathbf{X}_{\ell+c-1})), \quad (6)$$

where the subscript index is modulo $c$. We start with the first convolution filter $\mathbf{F}_0 = \{\mathbf{F}_{0\ell} \in \mathbb{R}^{f_h \times f_w}\}_{\ell=0}^{c-1}$ while taking into account the first $h \cdot w$ plaintext slots. The homomorphic convolution consists of two steps: (i) Extra-rotation: the input ciphertext is rotated by multiples of $h \cdot w$, and this action corresponds to a kernel-wise process and (ii) Intra-rotation: at each rotating position $\ell$, we perform a single-channel convolution on the rotated ciphertext of the input channel $\mathbf{X}_\ell$ with the kernel $\mathbf{F}_{0\ell}$, as in Equation (5). We repeat the process for all the rotating positions and sum up the results to generate a single-output channel. Only the first $h \cdot w$ entries for the convolution with $\mathbf{F}_0$ were used. If $h \cdot w \cdot c$ is less than the maximum length of plaintext vectors from the encryption

parameter setting, then we can pack together $c$ distinct kernels of the feature maps in plaintext slots and perform $c$ ordinary convolutions simultaneously in a SIMD manner without additional cost; the resulting ciphertext represents $c$ output channels stacked together.

In general, a convolutional layer is parameterized by $c_{in}$ and $c_{out}$, which indicate the number of input channels and output channels. We use $\bar{c}_{in}$ and $\bar{c}_{out}$ to denote the numbers of the input channels and output channels to be packed into a single ciphertext, respectively. Then the number of input ciphertexts and output ciphertexts are $n_{in} = \lceil c_{in}/\bar{c}_{in} \rceil$ and $n_{out} = \lceil c_{out}/\bar{c}_{out} \rceil$, respectively. Suppose that a ciphertext $ct_i$ represents the tensor input obtained by extracting from the $(\bar{c}_{in} \cdot (i-1)+1)$-th channel to $((\bar{c}_{in} \cdot i))$-th channel. For $j = 1, 2..., n_{out}$, the multi-channel convolution of the $j$-th output block can be securely computed by

$$\text{HE} - \text{Conv}_j(ct_1, \ldots, ct_{n_{in}}, \{pt.F_{i,j,k,\ell}\}) = \sum_{1 \le i \le n_{in}} \sum_{0 \le \ell < \bar{c}_{in}} \text{HE} - \text{Conv}(\rho^{\text{dist} \cdot \ell}(ct_i), \{pt.F_{i,j,k,\ell}\}) \tag{7}$$

$$= \sum_{1 \le i \le n_I} \sum_{0 \le \ell < \bar{c}_{in}} \sum_{|k| \le f} \text{MultPlain}(\rho^{r_k + \text{dist} \cdot \ell}(ct_i), pt.F_{i,j,k,\ell}), \tag{8}$$

where $f$ is defined as $(1 + 2 \cdot \lfloor f_h/2 \rfloor) \cdot (1 + 2 \cdot \lfloor f_w/2 \rfloor)$, pt. $F_{i,j,k,\ell}$ are plaintext polynomials that have the weights of the filters in appropriate locations, dist indicates the distance of two adjacent channels over plaintext slots (e.g., dist $= h \cdot w$ in Fig. 6b), $r_k$ denotes a rotation amount for the ordinary convolution (e.g., $r_k = u \cdot w + v$ in Equation (5)). To be precise, the output ciphertext represents from the $(\bar{c}_{out} \cdot (j-1)+1)$-th output channel to the $((\bar{c}_{out} \cdot j))$-th output channel, so that the output channels are stored in row-major order over all the ciphertexts.

## Fast homomorphic convolutions

We propose a fast homomorphic convolution operation that uses the merge-and-conquer method, which extensively uses components that have non-valid values of input ciphertexts to exploit the SIMD parallelism of computation. We take into account convolution operations that take as input punctured ciphertexts, which is a typical case in CNN. For a convolutional layer with $c_{out}$ feature maps of size $c_{in}$, we achieve this in three steps. We assume that each input ciphertext has valid values of $c$ input channels. (i) We multiply the output ciphertexts of the pooling layer by a constant zero-one plaintext vector to annihilate the junk entries marked # in Fig. 6c. As mentioned above, these non-valid entries are derived from rotations for the pooling operation. We then rotate each ciphertext by an appropriate amount and sum up all the resulting ciphertexts to obtain a ciphertext that contains all the valid entries of the response maps of the pooling. This procedure can be seen as a homomorphic concatenation of the sparsely packed ciphertexts of output channels from the pooling layer. We define the load number $n_P$ as the number of ciphertexts to fit into a single ciphertext in the preprocessing step. Then we need $n_P$ plaintext-ciphertext multiplications and $(n_P - 1)$ rotations to bring them together. Now each output ciphertext contains $(c \cdot n_P)$ valid values. (ii) We then conduct the ordinary homomorphic convolution with $c_{in}/(c \cdot n_P)$ input ciphertexts, in which each ciphertext contains $(c \cdot n_P)$ intermediate convolution results. (iii) We sum these results across plaintext slots to get $c$ output channels from $(c \cdot n_P)$ intermediate results. It can be done by doing precisely the opposite of the first step, that is, performing $(n_P - 1)$ rotations with the same amounts of the first step in the reverse direction. Furthermore, we can reduce the number of rotations down to $\lceil \log n_P \rceil$ rotations by accumulating them with recursive rotate-and-sum operations (Supplementary Note 2).

## Non-convolutional layers

Previous studies[24,28] collapsed adjacent linear layers such as convolution and pooling layers. We observe that the following layers can be collapsed while maintaining the same network structure: addition of a bias term in convolution operation, BN, polynomial activation, and scaling operation of the average pooling. We can adjust these parameters during the training phase, so they can be precomputed before secure inference. As a result, the collapsed layers become a polynomial evaluation per feature map, which applies to the elements of the same feature map in a SIMD manner. After feature extraction, the final $c_{in}$ outputs are fed into a FC layer with $c_{out}$ output neurons. Let $\mathbf{W}$ and $\mathbf{v}$ be the $c_{out} \times c_{in}$ weight matrix and length-$c_{in}$ data vector, respectively. The input vector $\mathbf{v}$ is split into sub-strings with the same length $c = 16$, i.e., it is given as multiple ciphertexts (each of which has $c$ values of the input vector in a sparse way). To align with this format, we split the original matrix $\mathbf{W}$ into $(c_{out} \times c)$-sized smaller blocks and perform computation on the sub-matrices. Consequently, the output ciphertext has $c_{out}$ predicted results (Supplementary Note 3).

## Data encryption

The CKKS cryptosystem supports homomorphic operations only on encrypted vectors, so an input tensor needs to be converted into such a plaintext format. Let $N_2 = N/2$, which is the maximum length of plaintext vector from the encryption parameter setting. We denote by $\text{PoT}(x)$ the smallest power-of-two integer that is greater than or equal to $x$. $J$ is the number of skeleton joints in each frame and $T$ is the number of frames of the skeleton sequence. Using the estimated skeleton joints of size $2 \times T \times J$, each 2D channel of size $T \times J$ is converted to a 1D vector in row-major order. Then it is zero-padded on the right to make the size of the vector as a power-of-two, so that $N_2$ is divisible by the vector size. One way to encrypt the input tensor is to generate a ciphertext that holds the concatenation of the two converted vectors. Alternatively, we stack as many copies of the input tensor as possible while interlacing the input channels, so that we can fully exploit the plaintext space for homomorphic computation. Afterward, we encrypt it as a fully packed ciphertext, then feed the generated ciphertexts into the CNN evaluation. We remark that if we do not pad with extra zero entries and make as many copies of the input, then the resulting plaintext vector has zeros in the last few entries, so those positions have different rotation results.

## Algorithmic and cryptographic optimizations

We employ the residue number system (RNS) variant of the CKKS scheme[43] to achieve efficiency of homomorphic operations. We first reformulate homomorphic convolution by applying the idea of the baby-step/giant-step algorithm[44]. Permutations on plaintext slots enable us to interact with values located in different plaintext slots; however, these operations are relatively expensive, so we aim to elaborate on the efficient implementation of Equation (8) to reduce the number of rotations by using the identity $\rho^{a+b} = \rho^a \circ \rho^b = \rho^b \circ \rho^a$ for any integers $a$ and $b$. (i) Full-step strategy: We precompute all the rotated ciphertexts of the form $\rho^{r_k + \text{dist} \cdot \ell}(ct_i)$ and perform plaintext-ciphertext multiplications by pt. $F_{i,j,k,\ell}$. (ii) Giant-step strategy: Equation (8) can be reformulated as

$$\sum_k \rho^{r_k} \left( \sum_{i,\ell} \text{MultPlain}(\rho^{\text{dist} \cdot \ell}(ct_i), \rho^{-r_k}(pt.F_{i,j,k,\ell})) \right). \tag{9}$$

This method is to precompute the rotated giant ciphertexts $\rho^{\text{dist} \cdot \ell}(ct_i)$'s for $i$ and $\ell$, perform plaintext-ciphertext multiplications, sum up the product results, and perform the evaluation of the rotation $\rho^{r_k}$. (iii) Baby-step strategy: The equation can be expressed as

$$\sum_\ell \rho^{\text{dist} \cdot \ell} \left( \sum_{i,k} \text{MultPlain}(\rho^{r_k}(ct_i), \rho^{-\text{dist} \cdot \ell}(pt.F_{i,j,k,\ell})) \right). \tag{10}$$

Therefore, one can precompute the rotated baby ciphertexts $\rho^{r_k}(ct_i)$'s for $i$ and $k$, aggregate the products, and perform the

evaluation of the rotation $\rho^{\text{dist}\cdot\ell}$. In practice, the evaluation strategies show different running time tendencies depending on the number of input/output ciphertexts (Supplementary Table 1 and Supplementary Fig. 2).

We adopt three cryptographic optimizations for homomorphic computation: (i) Hoisting optimization: One can compute the common part of multiple rotations on the same input ciphertext. We note that we can benefit from the hoisting optimization to reduce the complexity of multiple rotations on the same input ciphertext. That is, we can compute only once the common part that involves the computation of the number theoretic transformation (NTT) conversion on the input. As a result, the required number of NTT conversions can be reduced from $k$ to 1 if we use hoisting optimization on $k$ rotations of a ciphertext instead of applying each one separately. The hoisting technique is exploited for homomorphic convolution operations. (ii) Lazy-rescaling: Rescaling is not necessary after every multiplication. For instance, when evaluating Equation (8), we can first compute products between plaintext polynomials and ciphertexts, sum up all the resulting ciphertexts, and perform the rescaling operation only once to adjust the scaling factor of the output ciphertext. (iii) Level-aware model parameter encoding: When using plaintext polynomials of the trained model parameters, only a small subset of polynomial coefficients is needed for computation (Supplementary Fig. 1).

## Experimental setting
Our experiments were conducted on a machine equipped with an Intel Xeon Platinum 8268 2.9 GHz CPU with a 16-thread environment. Our source code is developed by modifying Microsoft SEAL version 3.4[45], which implements the RNS variant of the CKKS scheme. All experiments used encryption parameters to ensure 128 bits of security against the known attacks on the LWE problem from the LWE estimator[46] and HE security standard white paper[47] (Supplementary Table 2).

## Training
We used Stochastic Gradient Decent (SGD) optimizer with a mini-batch size of 64, a momentum of 0.9, and a weight decay of $5e^{-4}$ to train the model for 200 epochs. The initial learning rate was set to 0.05 with a decay of 0.1.

## Data preprocessing
The coordinate values for joints that were not detected or were detected with low probability were set as zero. For the frame selection mechanism, we calculate the Euclidean distance for the corresponding joint location for two consecutive frames. We calculate the mean of the distances to calculate the interchangeability score for the frames. If the score is below the predefined threshold of 5, the frame is dropped until we reach the required number of selected frames. This mechanism ensures that the action recognition network is independent of the frame per second (FPS) rate of the video camera.

First, the skeleton joints of each frame is encoded to 2D coordinates. Then, the joint location values are normalized separately for the two coordinates by applying the min-max normalization method. The normalization ensures that the action recognition network can work independently of body size or distance to the camera. Afterward, the coordinates of all joint coordinates in each frame are separately concatenated in a way that the spatial structure of each frame is represented as rows and the temporal dynamics across the frames in a video is encoded as changes in columns. Finally, 32 frames are selected to generate a 3D tensor of size $2 \times 32 \times 15$.

## Theoretical comparison to prior work
Throughput-optimized methods such as CryptoNets and nGraph-HE2 require $O(f_h \cdot f_w \cdot h \cdot w \cdot c_{\text{in}} \cdot c_{\text{out}})$ plaintext-ciphertext multiplications

to homomorphically evaluate a convolutional layer of kernel size $(f_h \times h_w)$ with $c_{\text{out}}$ feature maps on a $(c_{\text{in}} \times h \times w)$-sized input. In LoLa, the first convolutional layer is implemented using a restricted linear operation. To be precise, given a weight vector $\mathbf{w} = (w_j)$ of length $r$ and an input data vector $\mathbf{v} = (v_k)$, there exists a set of permutations $\sigma_i$ such that the $i$-th output of the linear transformation is $\sum_{1 \le j \le r} w_j v_{\sigma_i(j)}$. Therefore, the output can be computed using $r$ plaintext-ciphertext multiplications with $r$ ciphertexts of $(v_{\sigma_i(j)})$. In general, assuming that the entries of the data vector are encrypted as a single ciphertext, the network input is represented as $r = f_h \cdot f_w \cdot c_{\text{in}}$ ciphertexts to perform 2D convolutions using $r$ plaintext-ciphertext multiplications. The subsequent convolutional layers are represented as a series of dot-products between input neurons and one channel of size $f_h \cdot f_w \cdot c_{\text{in}}$, each requiring $O(\log_2(f_h \cdot f_w \cdot c_{\text{in}}))$ homomorphic operations. This process is repeated as many times as the number of output channels in the layer, so it imposes a complexity of $O(h \cdot w \cdot c_{\text{out}} \cdot \log_2(f_h \cdot f_w \cdot c_{\text{in}}))$. Meanwhile, Fast-HEAR requires $O(f_h \cdot f_w \cdot c_{\text{in}} \cdot c_{\text{out}})$ plaintext-ciphertext multiplications.

## Related work
In other recent work, the TFHE scheme[14] was used for secure neural network inference on Boolean circuits[48]. However, it is relatively slow for integer arithmetic, and is therefore not practically applicable in large neural networks for time-sensitive tasks. In the SHE system[32], the ReLU and max-pooling are expressed as Boolean operations and implemented by the TFHE homomorphic Boolean gates. Although SHE achieves state-of-the-art inference accuracy on the CIFAR-10 dataset, it requires thousands of seconds to make inference on an encrypted image. The most relevant studies are LoLa[28], CHET[49], and EVA[50], which use the ciphertext packing method to represent multiple values from network nodes as the same ciphertext. In LoLa, the convolutional layer is expressed as a restricted linear operation or matrix-vector multiplication, which requires a substantial number of rotations for an evaluation of convolution operations. In an orthogonal direction, CHET and EVA are FHE-based optimizing compilers to ease the task of making secure predictions by simplifying neural networks to homomorphic circuits. Their general-purpose solutions cannot fully take advantage of advanced techniques of FHE, and therefore may not be optimal for all tasks in either time or space. In contrast to their generalized approach, we come up with an efficient method to perform CNN evaluation by investigating the structure of CNN models and expressing required operations in an HE-compatible manner. In particular, our approach is efficient in computation complexity by exploiting the plaintext space and performing homomorphic convolutions in parallel.

## Reporting summary
Further information on research design is available in the Nature Research Reporting Summary linked to this article.

# Data availability
The ADL data are available from the J-HMDB (http://jhmdb.is.tue.mpg.de). The fall action class data are available from the URFD (http://fenix.univ.rzeszow.pl/mkepski/ds/uf.html) and Multicam (http://www.iro.umontreal.ca/labimage/Dataset/). The dataset used for pretrain is available at MPII Human Pose dataset (http://human-pose.mpi-inf.mpg.de). The raw data used for secure inference in this study are publicly available at https://github.com/K-miran/HEAR[51]. Source data are provided as a Source Data File. Source data are provided with this paper.

# Code availability
The software code of the secure CNN inference is publicly available at https://github.com/K-miran/HEAR[51].

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

## Acknowledgements

This work of M.K. and E.I. was supported by Basic Science Research Program through the National Research Foundation of Korea(NRF) funded by the Ministry of Education (No.2021R1C1C101017312). X.J. is CPRIT Scholar in Cancer Research (RR180012), and he was supported in part by Christopher Sarofim Family Professorship, UT Stars award, UTHealth startup, the National Institutes of Health (NIH) under award number R13HG009072 and R01AG066749-S1.

## Author contributions

All authors designed the secure action recognition scenario. M.K. and S.S. conceived the methodology. M.K., E.I., and S.S. implemented the software. M.K. conducted the benchmarking experiments and supervised the work. All authors wrote the manuscript and approved the final manuscript.

## Competing interests

The authors declare no competing interests.
