## [Peer Review File · Nature Communications]

Reviewers' Comments:

Reviewer #1:

Remarks to the Author:

Comments:

This paper combines machine learning techniques and Fully Homomorphic Encryption (FHE) to propose the first work to homomorphically evaluate neural networks for skeleton-based action recognition tasks on large-scale datasets. In essence, this paper presents a secure inference solution for a specific neural network based on FHE. To improve the efficiency, the authors make several algorithmic and cryptographic optimizations for the FHE cryptosystem CKKS. As a whole, the design is intuitional. It lacks evidence to demonstrate that this paper is superior to previous secure inference solutions. As far as I am concerned, the contribution of this paper is limited. Please see the detailed comments as follows.

- This paper explores CKKS to design a secure inference for neural networks and make some optimizations for CKKS. The idea is easy to come up with. Also, I fail to see evidence that shows the work outperforms priors solutions. Intuitively, it is not difficult to use the state-of-the-art secure inference solution to address skeleton-based action recognition tasks. So, it prevents me from recommending to accept this paper.
- The organization of this paper is not friendly for a reader. Why does the author separate the description of the chart from the chart? The organization of the current version is not smart to show the key idea and contribution of authors.
- Researchers have made so many efforts to secure inference of neural networks, however, this paper does not compare with the previous solutions. So, it fails to demonstrate the design of this paper is efficient.
- The framework of HEAR can be improved. HEAR requires a parameter server to distribute keys. In practice, it is not easy to set a trusted parameter server to distribute keys. Furthermore, the monitoring server in HEAR holds the private key. The monitoring server is responsible for generating and distributing keys which may be more practical.
- What is the data that the end-user submits to the cloud service provider? If the end-user submits feature vectors to the cloud service provider, the end-user holds a local model. In this case, why does she outsource the inference to the cloud service provider? If the end-user submits metadata to the cloud service provider, how does the end-user encrypt her data? Also, how does the cloud service provider extract encrypted feature vectors from encrypted data?
- The Fast-HEAR system lacks explanations. Also, some notations lack explanations, such as pk , sk , and evk .
- What is 2D/1D neural network model?
- As shown in Figure 3, please explain why there are two continuous convolution layers?
- From Figure 4, I see a high storage overhead for the solution proposed by this paper. So the authors should carefully optimize their design.

Reviewer #2:

Remarks to the Author:

Authors have proposed a synergetic system that is a skeleton-based privacy-preserving home activity recognition system using proposed a homomorphic encryption method named HEAR for securing the privacy of the users when the data is being transmitted over the network. The system never sends an image or video of the person instead it only sends encrypted skeleton structure for the corresponding pose. Authors claim the system is the first attempt to use homomorphic methods to evaluate NN for skeleton-based action recognition.

In general, the manuscript is written well and has merit in terms of methodology. The proposed method outperformed the existing method LOLA significantly inters memory usage, runtime. However, the manuscript is missing important parts in comparison methods with sota methods. It is not acceptable for publication in this current stage. Please make the revision according to the comments below:

1. Authors say the HEAR method is evaluated on three well-known public datasets, J-HMDB, URFD, and Multicam. It is unclear how the model performed on each dataset, there is no clear description of them. Please revise and make the clear.

2. For performance evaluation, authors have chosen specificity, sensitivity (recall), accuracy. It is well known that F1-score is suitable to evaluate model performance on the multi-labeled imbalanced datasets. It is required to add precision and F1-score in the metrics.
3. Since the HEAR and Fast-HEAR comparison is made on the runtime and memory usage. It is fair and informative to provide the performance evaluation comparison on the two methods.
4. Moreover, it is required to provide performance evaluation on Lola as it is their only method compared with.
5. As a comparison with the sota models, authors have used Lola. However, it is required to provide at least 3 sota methods unless there are no other methods available. However, there are several methods such as nGraph-HE, nGraph-HE2, Chameleon, Gazelle etc. in the similar task. Or at least, the authors should give a valid explanation why they have provided a single method as a comparison.

Author’s Response to Reviewers’ Comments of NCOMMS-21-48984-T

Miran Kim^{1,2,*}, Xiaoqian Jiang³, Kristin Lauter⁴, Elkhan Ismayilzada¹, and Shayan Shams^{3,*}

¹Department of Computer Science and Engineering, Ulsan National Institute of Science and Technology, Ulsan, 44919, Republic of Korea.

²Graduate School of Artificial Intelligence, Ulsan National Institute of Science and Technology, Ulsan, 44919, Republic of Korea.

³Center for Secure Artificial intelligence For hEalthcare (SAFE), School of Biomedical Informatics, University of Texas Health Science Center, Houston, TX, 77030, USA.

⁴Facebook AI Research (FAIR), Seattle, WA, USA.

We sincerely thank the reviewers and the editor for their constructive comments. We have addressed all of the comments and modified the paper accordingly. Our detailed responses and description of the revisions to each comment are below. We have highlighted the edited places in the manuscript with a **yellow highlight** to make the revised portions of the manuscript clear to the reviewers.

Response to Reviewer 1’s Comments

Reviewer 1:

This paper combines machine learning techniques and Fully Homomorphic Encryption (FHE) to propose the first work to homomorphically evaluate neural networks for skeleton-based action recognition tasks on large-scale datasets. In essence, this paper presents a secure inference solution for a specific neural network based on FHE. To improve the efficiency, the authors make several algorithmic and cryptographic optimizations for the FHE cryptosystem CKKS. As a whole, the design is intuitional. It lacks evidence to demonstrate that this paper is superior to previous secure inference solutions. As far as I am concerned, the contribution of this paper is limited. Please see the detailed comments as follows.

Authors’ Response: We sincerely thank the reviewer for the positive and constructive comments. We have now updated the Subsection named “Comparison to prior work” with additional information to demonstrate that our solutions achieve state-of-the-art inference latency and throughput over previous methods.

Major Comment 1.1: *This paper explores CKKS to design a secure inference for neural networks and make some optimizations for CKKS. The idea is easy to come up with. Also, I fail to see evidence that shows the work outperforms priors solutions. Intuitively, it is not difficult to use the state-of-the-art secure inference solution to address skeleton-based action recognition tasks. So, it prevents me from recommending to accept this paper.*

Researchers have made so many efforts to secure inference of neural networks, however, this paper does not compare with the previous solutions. So, it fails to demonstrate the design of this paper is efficient.

Authors’ Response: We agree with the reviewer’s concern about the necessity of comparison with prior work. We also thank the reviewer for pointing out the novelty of our work. The throughput-optimized methods such as CryptoNets and nGrarph-HE2 are to encrypt each node in the network as distinct ciphertexts and emulate unencrypted computation of neural network inference in the normal way. These methods are easy to implement, but they tend to suffer from high latency even for a single prediction and can lead to memory problems when applied to large-scale neural networks. The latency-optimized methods such as LoLa are to encrypt multiple values from network nodes as the same ciphertext, which is similar to ours. A convolutional layer is expressed as either a restricted linear operation or a series of dot products. These simplifications lead to a substantial number of homomorphic operations over large-dimensional inputs, which is the case for wide networks. In contrast, we first formulated homomorphic convolution operations as FHE-compatible operations on packed ciphertexts. To increase SIMD parallelism of computation, we extensively used entries that have non-valid values of ciphertexts. Furthermore, we introduced a range of algorithmic and cryptographic optimizations tailored to

increase the speed and reduce the memory usage of the secure inference from the approximate HE cryptosystem (CKKS). These innovations enable a significant reduction in running time and memory usage for secure inference compared to nGraph-HE2 and LoLa. We have updated the Subsection “Comparison to prior work” to include the results of the existing methods on action recognition tasks. Additionally, we provided a theoretical comparison of computational costs of homomorphic convolutions in Fast-HEAR, nGraph-HE2, and LoLa in the Section Methods.

Major Comment 1.2: *The organization of this paper is not friendly for a reader. Why does the author separate the description of the chart from the chart? The organization of the current version is not smart to show the key idea and contribution of authors.*

Authors’ Response: We agree with the reviewer’s concerns about the current structure of this paper. We have now updated all the figures and tables with detailed descriptions for a better understanding. In particular, we have updated Fig. 1 to provide a workflow of our cloud-based action recognition protocol and to present the cryptographic key management. We also updated the Subsection “Innovation of HEAR” and Fig. 2 to show our method’s key ideas and contributions with a detailed explanation.

Major Comment 1.3: *The framework of HEAR can be improved. HEAR requires a parameter server to distribute keys. In practice, it is not easy to set a trusted parameter server to distribute keys. Furthermore, the monitoring server in HEAR holds the private key. The monitoring server is responsible for generating and distributing keys which may be more practical.*

Authors’ Response: We agree with the point raised by the reviewer. We updated the system model in which a trusted monitoring service provider generates and distributes HE keys. We provide a more detailed description of our system in Figure 1.

Major Comment 1.4: *What is the data that the end-user submits to the cloud service provider? If the end-user submits feature vectors to the cloud service provider, the end-user holds a local model. In this case, why does she outsource the inference to the cloud service provider? If the end-user submits metadata to the cloud service provider, how does the end-user encrypt her data? Also, how does the cloud service provider extract encrypted feature vectors from encrypted data?*

Authors’ Response: Our metadata is the feature vector extracted by the pose estimation algorithm. Then the extracted joint keypoints are encrypted by the data owner, and HE algorithms operate upon such encrypted metadata to predict actions. Our architecture is designed to maintain a separation between data owners and action recognition algorithm developers (or providers) for two reasons: (i) to protect the privacy of the data owner and (ii) to protect the intellectual properties of the algorithm developers. The algorithm developers do not need to send their code/executables to the data owner. They can keep improving developed algorithms and investigating new algorithms (not only just for fall detection), which can run on the same encrypted metadata sent by the data owners for multiple tasks.

In reality, different health-related events bear out different weights for each individual. A client (especially an elderly one who has comorbidities) may subscribe to multiple machine learning tasks; if so, they must be deployed across multiple neural networks and managed by the backend algorithms simultaneously, which poses non-trivial computational challenges to edge devices. In a cloud-based outsourced scenario, the user only needs to encrypt the data once before outsourcing them, and the encrypted data can be used for different tasks. This characteristic eliminates the need to reprogram the end devices whenever the model providers tweak neural network models, so the overhead of end-users is significantly reduced. As a result, our solution can support multiple concurrent and heterogeneous tasks in elastic cloud computing with mitigated privacy risks for end-users. Therefore, by secure outsourcing with HE, the architecture allows us to build a secure and privacy-preserving ecosystem between algorithm developers and data owners.

From a practical perspective, HE algorithms are desired to compute low-degree polynomials. However, advanced joint keypoint detection algorithms like HRnet cannot be represented by low-degree polynomials. It is possible to convert such algorithms to a lower-degree counterpart, but that would be out of the scope of this paper.

Minor Comment 1.1: *The Fast-HEAR system lacks explanations. Also, some notations lack explanations, such as pk , sk , and evk .*

Authors' Response: We thank the reviewer for pointing out the missing details and clarity of the presentation. We have updated Figure 1 to have a detailed explanation of the cryptographic keys and their usage in the protocol.

Minor Comment 1.2: *What is 2D/1D neural network model?*

Authors' Response: We consider two CNN models depending on the shape of input neurons and the movement of kernels for convolution operations. If the kernel slides along two dimensions across the data, it is referred to as 2D convolutions and the network with 2D convolutions is called 2D-CNN. If the kernel slides along one dimension over the data, it is referred to as 1D convolutions and the network with 1D convolutions is called 1D-CNN. We updated the Subsection "Network architecture for action recognition" in Results to clarify the description of the underlying CNN models. We also included a detailed description of 1D/2D convolutional neural network architectures in Figure 3.

Minor Comment 1.3: *As shown in Figure 3, please explain why there are two continuous convolution layers?*

Authors' Response: We followed a common and standard pattern of blocks in designing and using convolutional neural networks architecture. The neural network consists of a number of convolutional layers, which are each followed by a batch normalization, an activation layer, and a downsampling layer. We first tried a neural network with two convolutional layers, but this model did not provide good classification performance to detect falls. Thus, we decided to target a neural network with three convolutional layers. We updated the Subsection "Network architecture for action recognition" in Results to clarify the architectures of convolutional neural networks.

Minor Comment 1.4: *From Figure 4, I see a high storage overhead for the solution proposed by this paper. So the authors should carefully optimize their design.*

Authors' Response: We carefully re-implemented the proposed protocols to use a better multi-threading approach and memory management. As a result, we could reduce the memory usage of homomorphic computations by up to 22% less space than the submitted version. Also, we achieved a speedup of 1.4x~1.9x for encoding model parameters or making a secure inference on networks. We have updated the runtime and memory requirements results in the Section "Result" (Figure 4).

Response to Reviewer 2's Comments

Reviewer 2:

Authors have proposed a synergetic system that is a skeleton-based privacy-preserving home activity recognition system using proposed a homomorphic encryption method named HEAR for securing the privacy of the users when the data is being transmitted over the network. The system never sends an image or video of the person instead it only sends encrypted skeleton structure for the corresponding pose. Authors claim the system is the first attempt to use homomorphic methods to evaluate NN for skeleton-based action recognition. In general, the manuscript is written well and has merit in terms of methodology. The proposed method outperformed the existing method LOLA significantly inters memory usage, runtime. However, the manuscript is missing important parts in comparison methods with sota methods. It is not acceptable for publication in this current stage. Please make the revision according to the comments below:

Authors' Response: We thank the reviewer for pointing out the missing comparisons with prior work. We updated the Subsection "Comparison to prior work" to include the benchmark results of the state-of-the-art nGraph-HE2 and LoLa methods in terms of running time, memory usage, and classification performance. We demonstrate the efficiency of our secure neural networks predictions by enabling a 613x latency speedup over LoLa and achieving an average of 3.1x increase in inference over nGraph-HE2 (Fig. 5c). Here, our elaborated and fine-tuned solution shows state-of-the-art

inference latency and throughput over previous methods. We have the Subsection “Comparison to prior work” to discuss the results of two previous methods and show comparisons of the methods.

Major Comment 2.1: *Authors say the HEAR method is evaluated on three well-known public datasets, J-HMDB, URFD, and Multicam. It is unclear how the model performed on each dataset, there is no clear description of them. Please revise and make the clear.*

Authors’ Response: For each dataset, the skeleton joints are first arranged as a 3D tensor by concatenating the detected joint locations from 32 frames. The transformed samples from the three datasets are merged for analysis, and the merged dataset is randomly split into training and testing sets that contain 70% and 30%, respectively. We trained the convolutional neural networks on the training set, and the evaluation was performed on the testing dataset of 608 samples over neural networks. We updated the Subsections “Dataset” and “Data preprocessing” to provide a clear description of the dataset.

Major Comment 2.2: *For performance evaluation, authors have chosen specificity, sensitivity (recall), accuracy. It is well known that F1-score is suitable to evaluate model performance on the multi-labeled imbalanced datasets. It is required to add precision and F1-score in the metrics. Since the HEAR and Fast-HEAR comparison is made on the runtime and memory usage. It is fair and informative to provide the performance evaluation comparison on the two methods.*

Authors’ Response: We have now updated the performance evaluation results of all methods to include precision and F1-score (Figures 4f, 5f). We confirmed that HEAR and Fast-HEAR showed the same performance on the test set in all performance metrics (Figure 4f) Our secure inference solutions achieved the same performance on the test set as the unencrypted inference except for the classification accuracy (Figures 4g, 4h). We have updated the Subsection “Classification performance of private action recognition” to summarize the above points.

Major Comment 2.3: *Moreover, it is required to provide performance evaluation on Lola as it is their only method compared with.*

Authors’ Response: We have added the performance evaluation results of LoLa and nGraph-HE2 in Figure 5f. Both secure inference solutions have the same multiplicative circuit depth as our methods, so we set the same encryption parameters as ours. As a result, they achieved the same classification performance on the test set as our methods. We updated the Subsection “Comparison to prior work” with this result.

Major Comment 2.5: *As a comparison with the sota models, authors have used Lola. However, it is required to provide at least 3 sota methods unless there are no other methods available. However, there are several methods such as nGraph-HE, nGraph-HE2, Chameleon, Gazelle etc. in the similar task. Or at least, the authors should give a valid explanation why they have provided a single method as a comparison.*

Authors’ Response: LoLa optimizes for latency at the cost of reduced throughput, while nGraph-HE2 optimizes for throughput at the cost of high latency. We have updated the Section “Results” to include their experimental results and compare them with our methods.

Other approaches are available for privacy-preserving deep learning prediction that uses multi-party computation (MPC) and their combinations with HE (e.g., Chameleon, Gazelle, and Minionn). These methods provide good latency but assume the tolerance of intensive communication overhead, which is not feasible in practice. Most of all, their systems are interactive, so data providers should stay online during the entire protocol execution, and it is difficult to operate in reality. In our case, multiple values from input are encrypted as a single ciphertext and it is enough to be transmitted once before secure computation. Therefore, our solution is asymptotically more efficient in communication than those other approaches. Furthermore, a service provider performs a large amount of work, and a client does not need to be involved in the computation. We have added a detailed discussion about the differences between our method and MPC-based approaches.

Reviewers' Comments:

Reviewer #1:

None

Reviewer #2:

Remarks to the Author:

Despite the author's efforts in answering my comments, there is still a major concern.

I think, HEAR system contributes to the literature in terms of cloud-based secure privacy-preserved action recognition in the smart homes but not in the action recognition itself. The research work would have a substantial contribution if the HEAR system outperformed the sota action recognition methods on the 3 public datasets separately in terms of efficiency and accuracy while performing the encrypted recognition. The main issue with the current results is not comparable in terms of action recognition because the authors have selected certain action classes from 3 public datasets to make the dataset. Yet, the authors claim that they have achieved high-accurate action detection at a low cost in the abstract. Due to this major concern, I can not accept the manuscript for publication. Authors either provide new results to meet the current claim or change the claim to meet the current results.

Some minor comments:

1. Make sure to use the same format "(sec)" or "seconds" in the whole content. Look at the sentence "HEAR enables a single prediction in 7.1 seconds 54 (sec) on average over a 2D CNN model for action recognition tasks, while achieving 86.21% sensitivity and 99.14% specificity 55 in detecting falls." in the introduction.
2. Authors should also clarify the total throughput including the pose estimation + encryption, evaluation, and recognition with respect to the computing power in the appropriate place. In my understanding, the authors didn't include the pose estimation inference time. Otherwise, it is misleading to the readers.
3. Seems like Fig4.f and Fig5.f have some incorrect numbers. It is highly unlikely that the measures are the same. Please check if there is something wrong.

Author's Response to Reviewers' Comments of NCOMMS-21-48984A

Miran Kim^{1,*}, Xiaoqian Jiang², Kristin Lauter³, Elkhan Ismayilzada⁴, and Shayan Shams^{2,*}

¹Department of Mathematics, Hanyang University, Seoul, 04763, Republic of Korea.

²Center for Secure Artificial intelligence For hEalthcare (SAFE), School of Biomedical Informatics, University of Texas Health Science Center, Houston, TX, 77030, USA.

³Facebook AI Research (FAIR), Seattle, WA, USA.

⁴Department of Computer Science and Engineering, Ulsan National Institute of Science and Technology, Ulsan, 44919, Republic of Korea.

We really appreciate the Editor and the Reviewer's comments as well as constructive suggestions and have revised the manuscript accordingly. We have highlighted the edited places in the manuscript with a **yellow highlight** to make the revised portions of the manuscript clear to the reviewers.

Response to Reviewer 2's Comments

Reviewer 1:

Despite the author's efforts in answering my comments, there is still a major concern. I think, HEAR system contributes to the literature in terms of cloud-based secure privacy-preserved action recognition in the smart homes but not in the action recognition itself. The research work would have a substantial contribution if the HEAR system outperformed the sota action recognition methods on the 3 public datasets separately in terms of efficiency and accuracy while performing the encrypted recognition. The main issue with the current results is not comparable in terms of action recognition because the authors have selected certain action classes from 3 public datasets to make the dataset. Yet, the authors claim that they have achieved high-accurate action detection at a low cost in the abstract. Due to this major concern, I can not accept the manuscript for publication. Authors either provide new results to meet the current claim or change the claim to meet the current results.

Authors' Response: We agree with the reviewer's concern about our claim. We have updated the manuscript to change the claim to meet the current results.

Minor Comment 1.1: *Make sure to use the same format "(sec)" or "seconds" in the whole content. Look at the sentence "HEAR enables a single prediction in 7.1 seconds 54 (sec) on average over a 2D CNN model for action recognition tasks, while achieving 86.21% sensitivity and 99.14% specificity 55 in detecting falls." in the introduction.*

Authors' Response: We updated the manuscript to use the same format for reporting the timing results.

Minor Comment 1.2: *Authors should also clarify the total throughput including the pose estimation + encryption, evaluation, and recognition with respect to the computing power in the appropriate place. In my understanding, the authors didn't include the pose estimation inference time. Otherwise, it is misleading to the readers.*

Authors' Response: We agree with the reviewer's concern about the confusion of the total throughput. In our system, it takes around 94 milliseconds to extract 15 joint locations for each frame on a V100 GPU. So, we can detect skeleton joints from 32 frames of the generated clips within 3.008 seconds to generate a 3D tensor. We have updated the Subsection "Dataset" to clarify this result.

Minor Comment 1.3: *Seems like Fig4.f and Fig5.f have some incorrect numbers. It is highly unlikely that the measures are the same. Please check if there is something wrong.*

Authors' Response: We have checked that all the numbers in Fig4.f and Fig5.f are correct. As discussed in the Subsection "Classification performance", our secure neural network inference systems show an accuracy degradation in some neural nets but achieve the same classification performance of fall detection (sensitivity, specificity, precision, and F1-score) as the unencrypted computation. These results indicate that they are enough to distinguish between falls and non-falls.